# Glycerol suppresses glucose consumption in trypanosomes through metabolic contest

Stefan Allmann[1,2☯], Marion Wargnies[1,2☯], Nicolas Plazolles[1☯], Edern Cahoreau[3,4], Marc Biran[2], Pauline Morand[2], Erika Pineda[1], Hanna Kulyk[3,4], Corinne Asencio[1], Oriana Villafraz[1], Loïc Rivière[1], Emmanuel Tetaud[1], Brice Rotureau[5], Arnaud Mourier[6], Jean-Charles Portais[3,4,7], Frédéric Bringaud[1,2]*

**1** Microbiologie Fondamentale et Pathogénicité, UMR 5234, Bordeaux University, CNRS, Bordeaux, France, **2** Centre de Résonance Magnétique des Systèmes Biologiques, UMR 5536, Bordeaux University, CNRS, Bordeaux, France, **3** Toulouse Biotechnology Institute, Université de Toulouse, CNRS, INRA, INSA, Toulouse, France, **4** MetaToul–MetaboHUB, Toulouse, France, **5** Trypanosome Transmission Group, Trypanosome Cell Biology Unit, Department of Parasites and Insect Vectors, INSERM U1201, Institut Pasteur, Paris, France, **6** Institute of Biochemistry and Genetics of the Cell (IBGC), CNRS, Bordeaux University, Bordeaux, France, **7** STROMALab, Université de Toulouse, INSERM U1031, EFS, INP-ENVT, UPS, Toulouse, France

☯ These authors contributed equally to this work.
* frederic.bringaud@u-bordeaux.fr

**Data Availability Statement:** All relevant data are within the paper and its Supporting Information files.

## Abstract

Microorganisms must make the right choice for nutrient consumption to adapt to their changing environment. As a consequence, bacteria and yeasts have developed regulatory mechanisms involving nutrient sensing and signaling, known as "catabolite repression," allowing redirection of cell metabolism to maximize the consumption of an energy-efficient carbon source. Here, we report a new mechanism named "metabolic contest" for regulating the use of carbon sources without nutrient sensing and signaling. *Trypanosoma brucei* is a unicellular eukaryote transmitted by tsetse flies and causing human African trypanosomiasis, or sleeping sickness. We showed that, in contrast to most microorganisms, the insect stages of this parasite developed a preference for glycerol over glucose, with glucose consumption beginning after the depletion of glycerol present in the medium. This "metabolic contest" depends on the combination of 3 conditions: (i) the sequestration of both metabolic pathways in the same subcellular compartment, here in the peroxisomal-related organelles named glycosomes; (ii) the competition for the same substrate, here ATP, with the first enzymatic step of the glycerol and glucose metabolic pathways both being ATP-dependent (glycerol kinase and hexokinase, respectively); and (iii) an unbalanced activity between the competing enzymes, here the glycerol kinase activity being approximately 80-fold higher than the hexokinase activity. As predicted by our model, an approximately 50-fold down-regulation of the GK expression abolished the preference for glycerol over glucose, with glucose and glycerol being metabolized concomitantly. In theory, a metabolic contest could be found in any organism provided that the 3 conditions listed above are met.

**Funding:** FB's team is supported by the Centre National de la Recherche Scientifique (CNRS, https://www.cnrs.fr/) (financial support for consumables and salary of permanent positions), the Université de Bordeaux (https://www.u-bordeaux.fr/) (financial support for consumables and salary of permanent positions), the Agence National de Recherche (ANR, https://anr.fr/) through the grants GLYCONOV (grant number ANR-15-CE15-0025-01) and ADIPOTRYP (grant number ANR19-CE15-0004-01) (financial support for consumables and PM and EP salary) and the Laboratoire d'Excellence (https://www.enseignementsup-recherche.gouv.fr/cid51355/laboratoires-d-excellence.html) through the LabEx ParaFrap (grant number ANR-11-LABX-0024) (financial support for consumables and SA salary), the ParaMet PhD programme of Marie Curie Initial Training Network (https://ec.europa.eu/research/mariecurieactions/) (FP7-PEOPLE-2011-ITN-290080) (financial support for consumables and MW salary) and the "Fondation pour le Recherche Médicale" (FRM, https://www.frm.org/) ("Equipe FRM", grant n°EQU201903007845) (financial support for consumables and EP salary). BR is supported by and the Institut Pasteur (financial support for consumables and salary of permanent positions). JCP's team from Metabolomics & Fluxomics facilities (Toulouse, France, http://www.metatoul.fr) is supported by the Agence National de Recherche (ANR, https://anr.fr/) (grant MetaboHUB-ANR-11-INBS-0010) (financial support for consumables and salary of permanent positions).

**Competing interests:** The authors have declared that no competing interests exist.

**Abbreviations:** 1H-NMR, proton nuclear magnetic resonance; BSF, bloodstream form; CFP, cyan fluorescent protein; F1,6BP, fructose 1,6-bisphosphate; FRET, fluorescence resonance energy transfer; G6P, glucose 6-phosphate; GK, glycerol kinase; Gly3P, glycerol 3-phosphate; GPDH, glycerol-3-phosphate dehydrogenase; HK, hexokinase; IC-HRMS, ion chromatography high-resolution mass spectrometry; PCF, procyclic form; PTS, phosphoenolpyruvate:sugar phosphotransferase system; RNAi, RNA interference; YFP, yellow fluorescent protein.

# Introduction

The diauxic growth observed in microorganisms consists of the sequential use of carbon sources when several are available, with the first one consumed, often glucose, being the one that ensures the highest growth rate. This concept emerged in the 1940s with the description in prokaryotes of preference for certain sugars, such as glucose over lactose or maltose, followed by the first description in 1964 of the phosphoenolpyruvate:sugar phosphotransferase system (PTS) [1]. PTS is a carbohydrate transport and phosphorylation system composed of 3 protein complexes that regulates numerous cellular processes by either phosphorylating target proteins or interacting with them in a phosphorylation-dependent manner [2]. The diauxic growth pattern also occurs in yeasts, which first consume glucose; then, the fermentative product ethanol is oxidized in a noticeably slower second growth phase, if oxygen is available [3]. In addition, the presence of glucose suppresses molecular activity of yeasts involved in the use of alternate carbon sources [4]. Whether a carbon source behaves as a preferred or non-preferred one is not defined by its chemical structure but by the rate at which it enters metabolism. The mechanisms by which repression is imposed are quite variable; however, they follow a general pattern, with complex sensory systems relying mostly on protein kinases and phosphatases [4–6]. These carbon "catabolite repression" processes prevent expression of enzymes for catabolism of less preferred carbon sources when the preferred substrate is present.

*Trypanosoma brucei* is a unicellular eukaryote that causes human African trypanosomiasis, also known as sleeping sickness [7]. Parasite transmission between mammals (bloodstream form [BSF] of *T. brucei*) is ensured by a hematophagous insect vector of the genus *Glossina* (tsetse fly). When grown *in vitro* in standard rich medium, the procyclic form (PCF) of *T. brucei*, present in the digestive tract of the insect vector, metabolizes glucose, which is converted by aerobic fermentation into partially oxidized end products, succinate, acetate, and alanine [8–10]. One unique particularity of trypanosome glycolysis is the occurrence of the first 6 glycolytic steps in specialized peroxisomes called glycosomes (see Fig 1A), while this pathway is cytosolic in all other eukaryotes [11]. No exchange of nucleotides has been described so far between the glycosomal and cytosolic compartments. Consequently, consumption and production of ATP are tightly balanced within the organelle [12].

Here we report a novel molecular mechanism for management of available resources, named "metabolic contest," that does not require complex sensory and signal transduction systems, as opposed to "catabolite repression." This is illustrated by the PCF trypanosomes, which prefer glycerol, a gluconeogenic carbon source, to glucose. The glycerol preference is due to the approximately 80-fold excess of glycosomal glycerol kinase (GK) activity (EC 2.7.1.30; the first step of glycerol assimilation) compared to glycosomal hexokinase (HK) activity (EC 2.7.1.1; the first glycolytic step), which compete for the same glycosomal pool of ATP (Fig 1A).

# Results and discussion

## Glycerol down-regulates glucose catabolism

The PCF of *T. brucei* catabolizes glucose and glycerol within glycosomes [13] (see Fig 1A). To determine their preferred carbon source, we first measured the consumption of glucose or glycerol by PCF trypanosomes maintained in culture in SDM79 medium supplemented with either glycerol or glucose or both. As expected, glycerol (compound with 3 carbons) was consumed by PCF faster than glucose (compound with 6 carbons). The rate of glycerol consumption was not affected by the presence of glucose. In contrast, the latter was not consumed as long as glycerol was present in the medium (Fig 1B). After glycerol exhaustion, glucose was consumed at a rate similar to under glucose-alone conditions. This absence of glucose

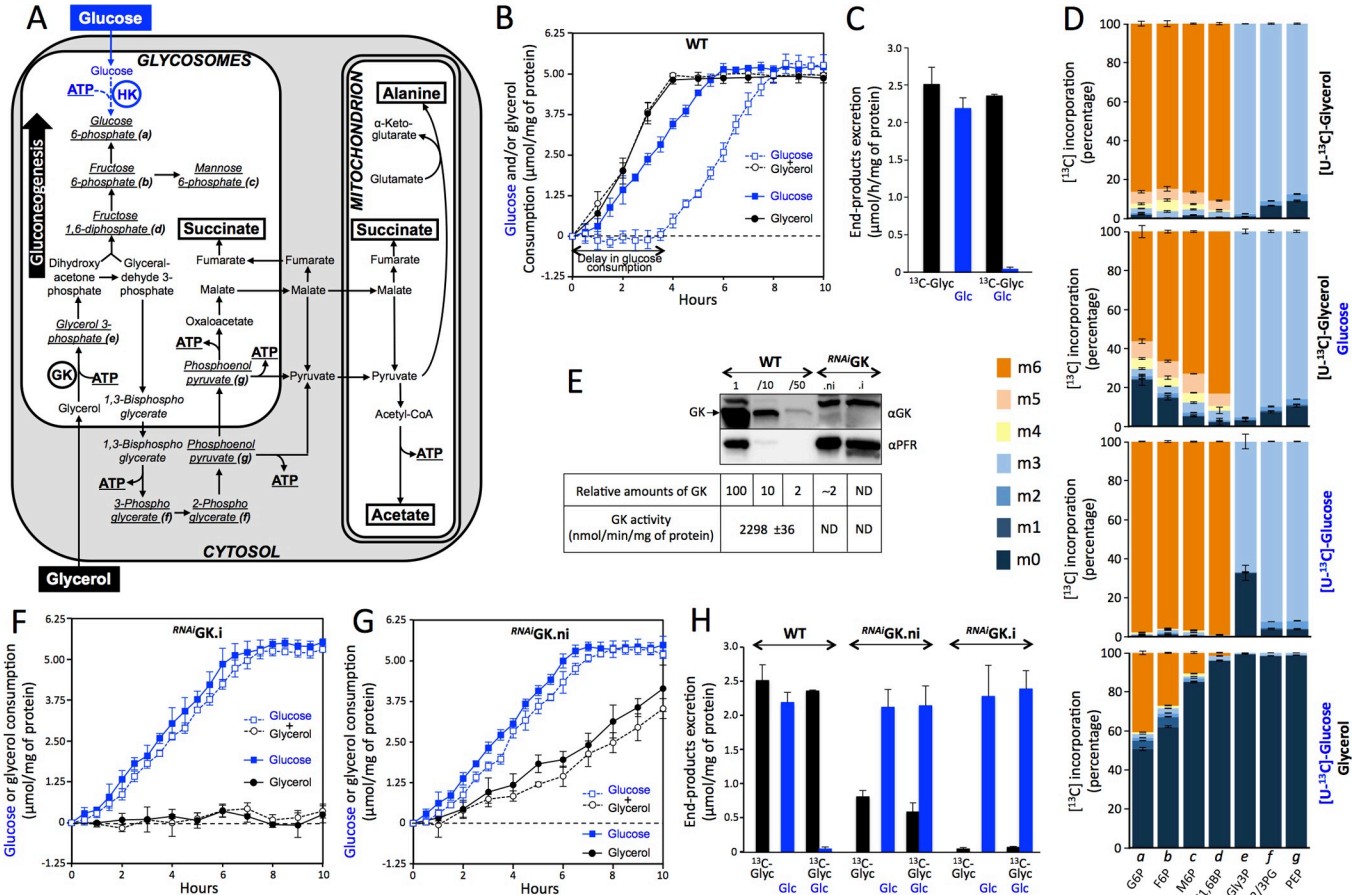

**Fig 1. Procyclic trypanosomes prefer glycerol to glucose.** (A) Schematic representation of glycerol (black) and glucose (blue) metabolism in procyclic form (PCF) trypanosomes. The metabolic end products are shown in rectangles, and metabolites analyzed by ion chromatography high-resolution mass spectrometry (IC-HRMS) are underlined and in italic (*a*–*g*). The ATP molecules consumed and produced by substrate-level phosphorylation are shown, as well as the enzymes hexokinase (HK) and glycerol kinase (GK). (B) Glucose and glycerol consumption by PCF trypanosomes incubated in glucose (2 mM), glycerol (2 mM) and glucose + glycerol (2 mM each) conditions. (C) Metabolic end products produced by PCF trypanosomes from [U-$^{13}$C]-glycerol ($^{13}$C-Glyc) and/or glucose (Glc), as measured by proton nuclear magnetic resonance ($^{1}$H-NMR) spectroscopy (the values are calculated from the data presented in S1 Table). (D) IC-HRMS analyses of intracellular metabolites collected from PCF trypanosomes after incubation with 2 mM [U-$^{13}$C]-labeled carbon sources in the presence or not of unlabeled carbon sources, as indicated on the right margin. The figure shows the proportion (%) of molecules having incorporated 0 to 6 $^{13}$C atoms (m0 to m6, color code indicated on the left margin). G6P (*a*), glucose 6-phosphate; F6P (*b*), fructose 6-phosphate; M6P (*c*), mannose 6-phosphate; F1,6BP (*d*), fructose 1,6-bisphosphate; Gly3P (*e*), glycerol 3-phosphate; 2/3PG, 2- or 3-phosphoglycerate (which are not undistinguished by IC-HRMS); PEP (*g*), phosphoenolpyruvate. (E) Western blot analysis of total protein extracts from the parental (wild-type [WT]) and tetracycline-induced (.i) or uninduced (.ni) $^{RNAi}$GK cell line probed with anti-GK (αGK) and anti-paraflagellar-rod (αPFR) immune sera. The table below the blots shows the relative levels of GK expression in $5 \times 10^{6}$ (1), $5 \times 10^{5}$ (/10) and $10^{5}$ (/50) parental cells, and $5 \times 10^{6}$ $^{RNAi}$GK.ni and $^{RNAi}$GK.i cells, as well as the corresponding GK activity. ND, not detectable. (F and G) Glucose and glycerol consumption by the (F) tetracycline-induced $^{RNAi}$GK ($^{RNAi}$GK.i) and (G) uninduced $^{RNAi}$GK ($^{RNAi}$GK.ni) mutant cell lines incubated in glucose (2 mM), glycerol (2 mM) and glucose + glycerol (2 mM each) conditions. (H) Production of metabolic end products by the parental (WT), $^{RNAi}$GK.ni, and $^{RNAi}$GK.i cell lines from [U-$^{13}$C]-glycerol ($^{13}$C-Glyc) and/or glucose (Glc), as measured by $^{1}$H-NMR spectroscopy (the values are calculated from the data presented in S1 Table). Data supporting the results described in this figure can be found at https://zenodo.org/record/5075637#.YORd2B069yA.

consumption in the presence of glycerol clearly showed a strong catabolic-repression-like effect exerted by glycerol on glucose. It is noteworthy that the consumption of glucose started as soon as glycerol was exhausted, with no delay.

To confirm this glycerol preference in PCF trypanosomes, we monitored the metabolic fate of uniformly $^{13}$C-labeled glycerol ([U-$^{13}$C]-glycerol) alone or in combination with equimolar amounts of unlabeled glucose. The analysis of metabolic end products by proton nuclear magnetic resonance ($^{1}$H-NMR) spectroscopy (Fig 1C; S1 Table) allowed determining the

respective contribution of [U-$^{13}$C]-glycerol (labeled compounds) and glucose (unlabeled compounds) [14–16]. Trypanosomes mostly excreted acetate and succinate from glycerol or glucose (see Fig 1A). The rate at which glycerol was converted into these compounds was only slightly modified by the presence of glucose. In contrast, the conversion of glucose into acetate and succinate was reduced by approximately 30-fold in the presence of [U-$^{13}$C]-glycerol. Moreover, the small production of lactate and alanine from glucose observed in the absence of glycerol was abolished in its presence. This significant reduction in the conversion of glucose into end products in the presence of glycerol was correlated with an approximately 20-fold decrease in glucose consumption in this experimental setup (see S1 Table), confirming that glucose metabolism was strongly down-regulated in the presence of glycerol.

Since production of glucose 6-phosphate (G6P) through gluconeogenesis is essential in the absence of glucose (see Fig 1A), we measured by ion chromatography high-resolution mass spectrometry (IC-HRMS) the incorporation of $^{13}$C label into glycolytic intermediates in PCF trypanosomes incubated with [U-$^{13}$C]-glycerol. In this experiment, most hexose phosphate glycolytic intermediates were fully $^{13}$C-labeled (88.2% ± 1.8% of total molecules on average) after 2 h of incubation with [U-$^{13}$C]-glycerol as the sole carbon source (Fig 1D). Addition of an equal amount of unlabeled glucose only slightly reduced $^{13}$C incorporation into hexose phosphates, with an average of 69.9% ± 9.7% fully $^{13}$C-labeled molecules (Fig 1D). To confirm this preference for glycerol over glucose, the equivalent experiment was performed with [U-$^{13}$C]-glucose (Fig 1D). Addition of an equal amount of unlabeled glycerol abolished incorporation of $^{13}$C from [U-$^{13}$C]-glucose into triose phosphates and fructose 1,6-bisphosphate (F1,6BP). The $^{13}$C incorporation into G6P was strongly reduced (40% ± 0.9% versus 98% ± 0.1% fully $^{13}$C-labeled molecules in the presence and absence of glycerol, respectively).

Altogether, these data demonstrate that PCF trypanosomes significantly prefer glycerol to glucose for the production of hexose phosphates, including the first glycolytic intermediate, i.e., G6P. These data also suggest that HK, which produces G6P from glucose, and/or the PCF glucose transporter (THT2), may be the target of the glycerol-induced down-regulation of glucose metabolism. As far as we are aware, PCF trypanosomes are the only extracellular microorganisms described to date showing a preference for glycerol over glucose. *T. brucei* PCF is also the only known glycolytic-competent lower eukaryote performing gluconeogenesis in the presence of glucose.

## Glycerol metabolism is critical for glucose catabolism repression

To further study glycerol metabolism in PCF trypanosomes, the expression of the first enzyme of the glycerol pathway was down-regulated by an RNA interference (RNAi) silencing approach simultaneously targeting the 5 tandemly arranged *GK* genes (Tb927.9.12550–Tb927.9.12630) under control of a tetracycline-inducible system. In the absence of tetracycline, the uninduced $^{RNAi}$GK ($^{RNAi}$GK.ni) cell line presented strong constitutive leakage of the RNAi silencing system, with a 50-fold reduction of GK protein content, reducing overall GK enzyme activity to an undetectable level (Fig 1E). The residual GK protein level could be further reduced after tetracycline induction ($^{RNAi}$GK.i). Thus, the direct involvement of GK in glycerol metabolism in these cells was determined by measuring glycerol consumption and release of metabolic end products under glycerol conditions (Fig 1F–1H; S1 Table). Both glycerol consumption (Fig 1F) and acetate/succinate production from glycerol metabolism (Fig 1H; S1 Table) were almost abolished in the $^{RNAi}$GK.i mutant, demonstrating that there is no alternative to GK for glycerol breakdown in PCF trypanosomes. Interestingly, the presence of glycerol did not affect glucose consumption by the $^{RNAi}$GK.i mutant (Fig 1F), indicating that the presence of glycerol in the medium is not *per se* responsible for glucose metabolism repression. In

other words, glycerol does not directly affect glucose uptake and metabolism, which implies that intracellular glycerol metabolism is required to repress glucose metabolism. It is also important to mention that replacing glucose by glycerol did not affect growth of the $^{RNAi}$GK.i mutant (S1 Fig), given that proline was the main carbon source used in these conditions, as in the insect vector midgut [8,10,17]. It is worth mentioning that knocking down the 10 identical *GK* genes is far easier than doing the alternative experiment consisting on knocking down/out the 3 different aquaglyceroporin genes (*AQP1–AQP3*) responsible for glycerol uptake in *T. brucei* [18].

The analysis of the $^{RNAi}$GK.ni cell line also provided relevant information regarding the unexpected role of GK activity in the preference for glycerol over glucose. First, the consumption of glycerol (Fig 1G) and its conversion into end products (Fig 1H; S1 Table) were reduced only by 3.5-fold and 3.1-fold, respectively, in the $^{RNAi}$GK.ni mutant as compared to the parental cells, while GK expression was approximately 50-fold down-regulated (Fig 1E). One can extrapolate from these data that a reduction of GK activity by at least 90% would not affect the glycerol metabolism flux, which would highlight a large excess of GK activity in PCF trypanosomes (in the range of 10-fold). Second, glucose metabolism was no longer repressed by glycerol in the $^{RNAi}$GK.ni cells, which consumed glucose at the same rate as the parental cells, without any glycerol-induced delay, although glycerol consumption remained constant over the 10 h of incubation (Fig 1G). This strongly suggests that the abolition of the glycerol preference in the $^{RNAi}$GK.ni cells could be the consequence of the 50-fold reduction of GK activity.

## The glucose catabolism repression is due to a large excess of GK activity

Interestingly, GK activity was approximately 80-fold higher than HK activity in total PCF extracts (Fig 2B), using the enzymatic assays described in Fig 2A. Since these 2 glycosomal enzymes can compete for the same ATP pool (glycosomal), we hypothesized that this significant difference in activity would favor glycerol metabolism and disfavor glucose metabolism, and would hence explain the unique repression of glucose by glycerol. To test this hypothesis, we measured HK activity in the presence or absence of glycerol under incubation conditions compatible with HK and GK activity. Importantly, the enzymatic assay included 0.6 mM ATP, which corresponds to the measured glycosomal concentration [19]. The presence of glycerol in the assay induced a 15-fold reduction of HK activity in the parental cell extracts, but not in the $^{RNAi}$GK.ni and $^{RNAi}$GK.i cell extracts (Fig 2B), which demonstrates that the conversion of glycerol into Gly3P, but not the presence of glycerol *per se*, inhibits HK activity and thus glucose metabolism. In contrast, GK activity was not impaired by the presence of glucose in the parental cell line extract (Fig 2B).

We took advantage of the fact that both the parental and $^{RNAi}$GK.i cell extracts displayed similar HK activity (see Fig 2B) to further characterize the GK-derived inhibition effect on HK activity as a function of the HK/GK activity ratio, by diluting the parental cell extract with different volumes of the $^{RNAi}$GK.i cell extract (Fig 2C). As expected, HK activity was equivalent in all the samples in the presence of 10 mM glucose (S2 Fig). However, the addition of 10 mM glycerol decreased HK activity, and this effect was dependent on the HK/GK ratio (Fig 2C). Indeed, a reverse correlation between HK and GK activity was observed (Fig 2D), which was consistent with our hypothesis that both enzymes are competing for the same ATP pool. To confirm that this inhibitory effect was due to GK activity rather than to any other activities or biochemical properties of the enzyme, GK and glycerol were replaced by acetate kinase and acetate in the same HK activity assay. As anticipated, HK activity was inhibited concomitantly with increasing amounts of acetate kinase (Fig 2E). Altogether, these data demonstrate that the

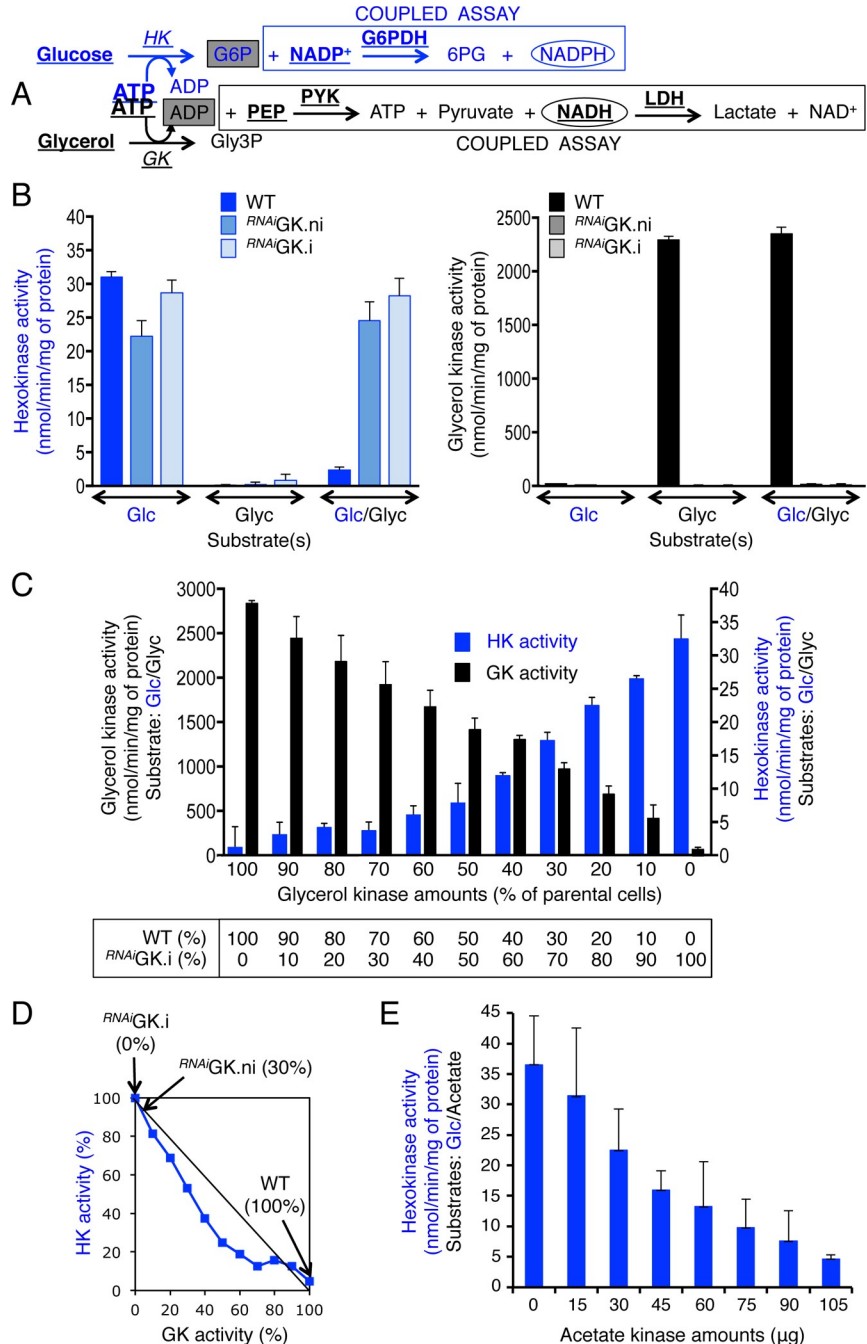

**Fig 2. The glycerol preference is the consequence of the high excess of GK activity.** (A) Enzymatic assays used for the quantification of hexokinase (HK) and glycerol kinase (GK) activity. The bold and underlined substrates and enzymes are included in the assay for production of NADPH (HK assay) and consumption of NADH (GK assay) that are detected by spectrometry at 350 nm. 6PG, 6-phosphogluconate; G6PDH, glucose-6-phosphate dehydrogenase; Gly3P, glycerol 3-phosphate; LDH, lactate dehydrogenase; PEP, phosphoenolpyruvate; PYK, pyruvate kinase. (B) HK (left panel) and GK (right panel) activity in total cell extracts (wild-type [WT], *RNAi*GK.i and *RNAi*GK.ni) determined in the presence of glucose (Glc), glycerol (Glyc) or equal amounts of glucose and glycerol (Glc/Glyc). (C) GK and HK activity in different combinations (indicated in the table below the graph) of total cell extracts from the parental (WT) and the *RNAi*GK.i cell lines. The amount of HK remained the same in all samples (see S2 Fig), while the amount of GK present in the parental samples was diluted with the GK-depleted *RNAi*GK.i samples. The HK and GK activity were determined in the presence of both glucose and glycerol, as in the Glc/Glyc condition (see [B]). (D) Expression of HK activity as a function of GK activity. The values in parentheses indicate the rate of glycerol consumption in the *RNAi*GK. ni and *RNAi*GK.i cells compared to the parental cells (100%) (see Fig 1F and 1G). (E) HK activity in the presence of 10

mM acetate and increasing amounts of acetate kinase. Data supporting the results described in this figure can be found at https://zenodo.org/record/5075637#.YORd2B069yA.

preference for glycerol over glucose is the consequence of a competition between HK and GK for their common substrate (ATP), which we named "metabolic contest."

## The GK/HK activity ratio is optimal for glycerol preference in procyclic trypanosomes cultured in glycerol-rich medium

As mentioned above, procyclic trypanosomes multiply in medium containing glycerol instead of glucose; however, all the biochemical experiments presented so far were performed on cells grown in standard glucose-rich conditions. Transferring the procyclic cells from glucose-rich to glycerol-rich conditions (without glucose) induced a 2.3-fold reduction of GK activity and a 1.4-fold increase of HK activity, with the GK/HK ratio reduced by 3.3-fold in glycerol-rich medium (Fig 3A). These changes in GK and HK activity were not observed under glucose/glycerol-depleted conditions, indicating that the presence of glycerol, but not the absence of glucose, is responsible for this adaptation. The glycerol-induced down-regulation of GK expression was confirmed by Western blotting, with a 3.5-fold reduction of the GK protein level 2 d after cell transfer to glycerol-rich conditions (Fig 3B). Interestingly, this phenomenon was reverted when replacing glycerol with glucose (Fig 3B), which suggests that a high level of GK expression is required for the cells to grow under glucose-rich conditions or that reduced GK expression is optimal for glycerol metabolism. To determine the effect of GK down-regulation on glycerol preference, we monitored by $^1$H-NMR spectrometry the metabolic fate of [U-$^{13}$C]-glycerol alone or in combination with equimolar amounts of unlabeled glucose. As expected, the rate of end product excretion from glycerol catabolism was not affected by the reduced expression of GK (Fig 3C). More importantly, the repression exerted by glycerol on glucose degradation was similar regardless of the growing conditions of the parasite (Fig 3C).

In order to modulate GK expression, a *GK* gene recoded for resisting RNAi silencing (*GKrec*), was introduced in the $^{RNAi}$GKcst cell line, in which the expression of the endogenous *GK* genes is constitutively down-regulated (Fig 3D). Because of the strong constitutive leakage of the RNAi silencing system, as observed above for the $^{RNAi}$GK.ni cell line (Fig 1E), the resulting uninduced $^{RNAi}$GKcst/$^{OE}$GKrec.ni cell line expressed GKrec with a GK activity lowered by 35% compared to that of the parental EATRO1125.T7T cells (Fig 3D). This reduced GK activity did not affect the preference for glycerol over glucose, as deduced from $^1$H-NMR analysis of excreted end products from glucose and/or glycerol metabolism (Fig 3C). As expected, the $^{RNAi}$GKcst/$^{OE}$GKrec.i cell line showed a 2-fold increase in GK activity upon tetracycline induction (Fig 3D) and maintained a preference for glycerol over glucose (Fig 3C). Altogether, these data show that a 3.3-fold reduction of the GK/HK activity ratio in the presence of glycerol does not affect the preference for glycerol over glucose. It is noteworthy that we also previously reported glycerol-induced down-regulation of GK expression (7-fold) in the BSF of *T. brucei* [20].

## GK and HK compete for glycosomal ATP

To further understand the mechanisms underlying the metabolic contest, we determined the ATP concentrations required to prevent HK activity in an excess of GK. The HK activity of trypanosome extracts was monitored over a period of incubation in the presence of glycerol and different amounts of ATP. In the presence of GK activity, HK activity was maintained until ATP concentrations reached the millimolar range (1.1 to 1.5 mM depending on the initial

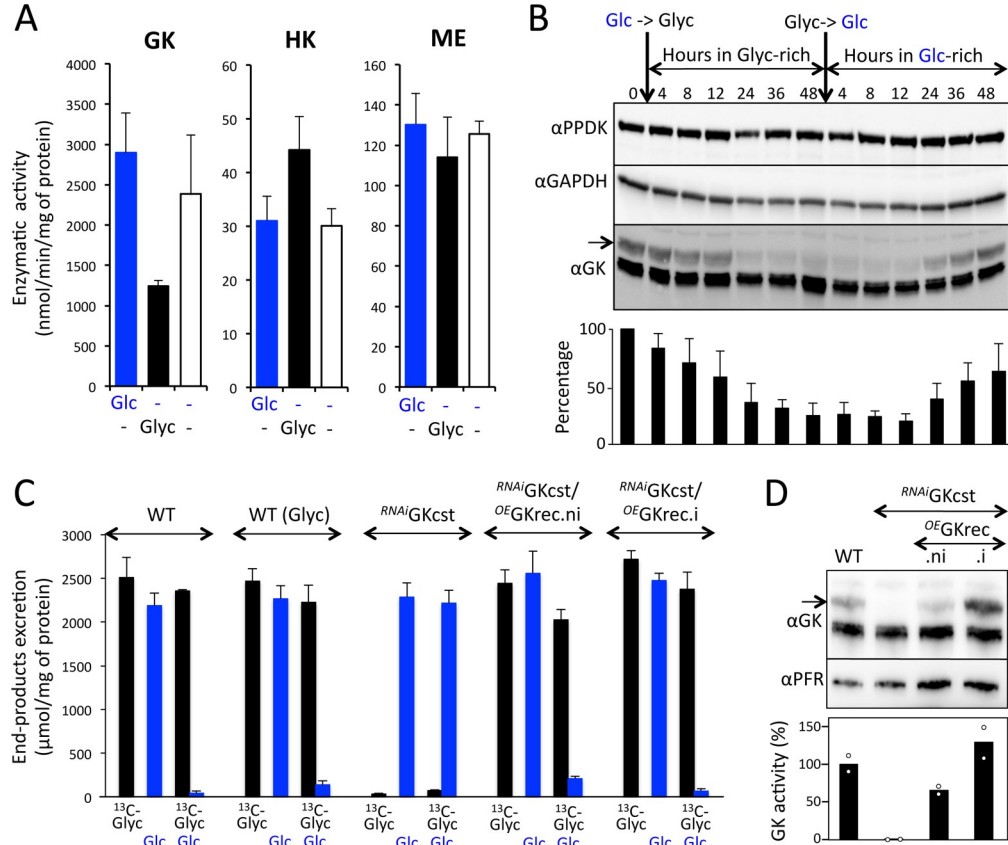

**Fig 3. Glycerol down-regulates GK expression but does not affect preference for glycerol over glucose.** (A) Glycerol kinase (GK), hexokinase (HK), and malic enzyme (ME) activity determined in total cell extracts of EATRO1125.T7T procyclic trypanosomes grown in glucose-rich (Glc/−), glycerol-rich (−/Glyc) or glucose/glycerol-depleted (−/−) conditions. (B) Western blot analysis of procyclic cells grown in glucose-rich medium (lane 0), then in glycerol-rich medium (in the absence of glucose and in the presence of N-acetyl-D-glucosamine) for 48 h, before reintroducing glucose (without glycerol and N-acetyl-D-glucosamine) for 48 h. The immune sera used against GK (αGK), pyruvate phosphate dikinase (αPPDK), and glyceraldehyde-3-phosphate dehydrogenase (αGAPDH) are indicated on the left margin. The bottom panel is a quantitative analysis of the GK signal indicated by an arrow (n = 4). (C) Metabolic end products of PCF trypanosomes from metabolism of [U-$^{13}$C]-glycerol ($^{13}$C-Glyc) and/or glucose (Glc) measured by proton nuclear magnetic resonance spectrometry (the values are calculated from the data presented in S2 Table). (D) Expression of the recoded (GKrec) and native GK in the wild-type (WT), $^{RNAi}$GKcst, and tetracycline-induced (.i) and uninduced (.ni) $^{RNAi}$GKcst/$^{OE}$GKrec cell lines monitored by Western blotting on total cell extracts using immune sera against GK (αGK) and paraflagellar rod (αPFR) as control (top panel), and GK activity assay normalized with malic enzyme activity and expressed as a percentage of activity in the WT cells (bottom panel, n = 2). Data supporting the results described in this figure can be found at https://zenodo.org/record/5075637#.YORd2B069yA.

amounts of ATP, i.e., 1.2 to 3 mM, respectively) (Fig 4A, top panel). However, in the presence of lower amounts of ATP such as 0.6 mM, no HK activity was detected, while GK was active for 1.5 min, until all ATP was consumed (Fig 4A, bottom panel). These data demonstrate that HK activity is inhibited by an excess of GK in the presence of physiological amounts of ATP (0.6 mM in glycosomes [19]); however, at higher ATP concentrations, HK is active until ATP levels drop below 1.5 mM. It is noteworthy that *T. brucei* HK and GK have the same affinity for ATP (Km = 0.28 mM and 0.24 mM, respectively [21,22]), suggesting that the preference for GK over HK at ATP concentrations below 0.6 mM is primarily due to the large excess of GK activity. This competition between HK and GK for glycosomal ATP was previously anticipated based on a kinetic model of trypanosome glycolysis [23].

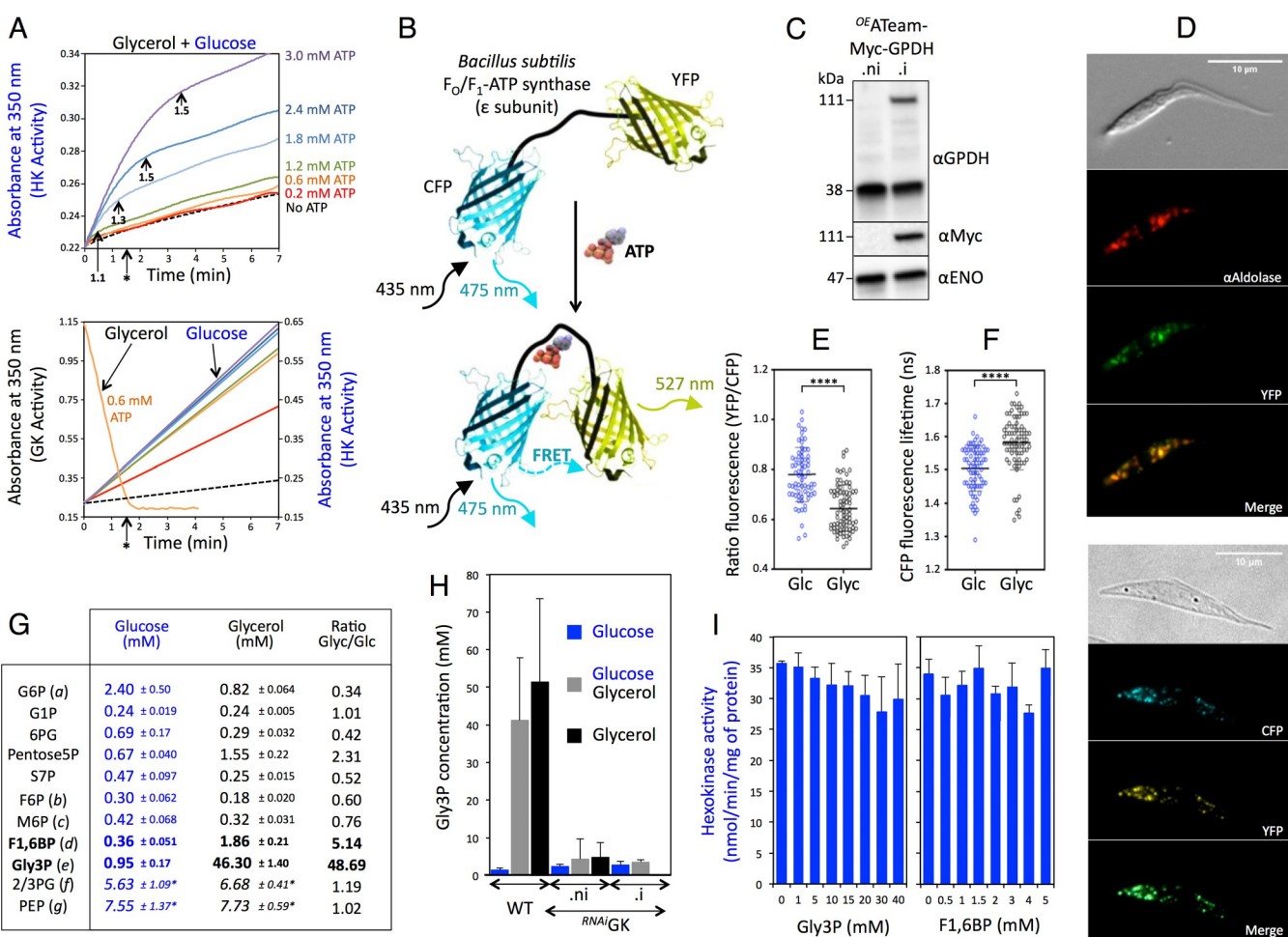

**Fig 4. Analysis of intracellular ATP and metabolites.** (A) The top panel shows hexokinase (HK) activity determined at 350 nm (NADPH production) over the incubation time of trypanosome extracts in the presence of 10 mM glucose and 10 mM glycerol and 0.2 to 3.0 mM ATP. The dashed lane corresponds to background HK activity measured without ATP. The arrows indicate the calculated ATP amounts (mM) remaining in the assay at the time of HK activity inhibition, taking into account glycerol kinase (GK) and HK activity. The asterisk indicates the time when 0.6 mM ATP is consumed by GK (deduced from the bottom panel). The bottom panel shows NADH consumption (GK activity) and NADPH production (HK activity) in the presence of 0.6 mM ATP (GK activity) or 0.2 to 3.0 mM ATP (HK activity). The dashed lane corresponds to HK activity measured without ATP. (B) Schematic drawing of the ATeam probe from [24]. Variants of cyan fluorescent protein (CFP; mseCFP) and yellow fluorescent protein (YFP; cp173-mVenus) were connected by the ε subunit of *Bacillus subtilis* $F_O F_1$-ATP synthase. In the ATP-free form (top), extended and flexible conformations of the ε subunit separate the 2 fluorescent proteins, resulting in a low fluorescence resonance energy transfer (FRET) efficiency. In the ATP-bound form, the ε subunit retracts to draw the 2 fluorescent proteins close to each other, which increases FRET efficiency. (C) The expression of ATeam-Myc-GPDH was controlled by Western blotting on total cell extracts of tetracycline-induced (.i) and uninduced (.ni) [OE]ATeam-Myc-GPDH cells using anti-GPDH (αGPDH) and anti-Myc (αMyc) immune sera, and as control anti-enolase (αENO) immune serum. (D) The subcellular localization of ATeam-Myc-GPDH was confirmed by immunofluorescence assays on the [OE]ATeam-Myc-GPDH.i cell line using an anti-aldolase immune serum as a glycosomal marker (top panel; the yellow YFP signal was converted to green to merge it with the red fluorescence corresponding to aldolase) or by observing the fluorescence activity of CFP and YFP (FRET) (bottom panel). (E) The ratio of YFP emission (FRET) and CFP emission after excitation at 435 nm in the [OE]ATeam-Myc-GPDH.i cell line incubated in the presence of glucose (Glc) or glycerol (Glyc) (mean ± SD, n = 2 independent experiments, ****p < 0.0001). (F) The CFP fluorescence lifetime of the same cell line incubated in the same conditions (mean ± SD, n = 3 independent experiments, ****p < 0.0001). (G) Intracellular concentrations of metabolites in procyclic form trypanosomes grown in the presence of 10 mM glucose or glycerol (letters in parentheses refer to Fig 1A). The concentrations of the 2 last metabolites (asterisk) cannot be calculated, and the ratio between the [12]C (sample metabolite) and [13]C (standard) area was considered. G6P, glucose 6-phosphate; G1P, glucose 1-phosphate; 6PG, 6-phosphogluconate; Pentose5P, pentose 5-phosphate (ribose 5-phosphate, xylulose 5-phosphate, and xylose 5-phosphate are not distinguished by ion chromatography high-resolution mass spectrometry); S7P, sedoheptulose 7-phosphate; M6P, mannose 6-phosphate; F6P, fructose 6-phosphate; F1,6BP, fructose 1,6-bisphosphate; Gly3P, glycerol 3-phosphate; 2/3PG, 2- or 3-phosphoglycerate; PEP, phosphoenolpyruvate. (H) Enzymatic determination of intracellular Gly3P concentration in the parental (wild-type [WT]), [RNAi]GK.ni and [RNAi]GK.i cell lines grown in 10 mM glucose (blue), 10 mM glycerol (black) or both (grey). The absence of detectable amounts of Gly3P in cellular extracts from the [RNAi]GK.i mutants maintained in glycerol (last column) is probably due to cell quiescence caused by the impossibility of this mutant to metabolize glycerol. The intracellular concentrations of metabolites are calculated with the assumption that the total cellular volume of $10^8$ cells is equal to 5.8 μL [27]. (I) Effect of increasing amounts of Gly3P and F1,6BP on HK activity determined in total extracts of procyclic form *T. brucei*. Data supporting the results described in this figure can be found at https://zenodo.org/record/5075637#.YORd2B069yA.

To estimate the impact of glycerol metabolism on glycosomal ATP levels, we used an ATP-specific fluorescence resonance energy transfer (FRET)-based indicator, named ATeam, that is composed of a bacterial $F_oF_1$-ATP synthase ε subunit sandwiched between cyan fluorescent protein (CFP) and yellow fluorescent protein (YFP) [24]. In the ATP-bound form, the ε sub-unit retracts to bring the 2 fluorescent proteins close to each other, which increases FRET efficiency and allows detection of changes in ATP level upon fluorescence quantification (Fig 4B). To focus on the glycosomal ATP levels, the ATeam cassette was fused to the N-terminal extremity of a Myc-tagged glycosomal protein containing a C-terminal peroxisomal targeting motif (PTS1), i.e., glycerol-3-phosphate dehydrogenase (GPDH; EC 1.1.1.8; Tb927.8.3530), which was recently used to target a cytosolic protein exclusively inside this organelle [25]. Upon tetracycline induction, the anti-GPDH and anti-Myc immune sera recognized a 111-kDa protein corresponding to the expected size of the ATeam-Myc-GPDH recombinant protein (Fig 4C). Immunofluorescence analyses showed that the ATeam-Myc-GPDH recombinant protein was located in glycosomes, as confirmed by the colocalization of the ATeam-Myc-GPDH detected signal (YFP) and the aldolase glycosomal marker (Fig 4D, top panel), as well as the colocalization of CFP and YFP FRET signals, forming glycosomal-like images (Fig 4D, bottom panel). FRET efficiency was significantly decreased in the $^{OE}$ATeam-Myc-GPDH.i cells incubated in the presence of glycerol compared to glucose conditions, which indicates that glycerol metabolism induced a reduction of the intraglycosomal ATP level compared to the standard glucose conditions (Fig 4E). This was confirmed by an increase in CFP fluorescence lifetime corresponding to a FRET decrease (Fig 4F). We concluded that this glycerol-induced reduced glycosomal ATP level favors glycerol preference.

To investigate whether parasite metabolic profiles were dependent on carbon source availability, we determined the absolute intracellular concentrations of metabolites by IC-HRMS by adding an internal standard ([U-$^{13}$C]-labeled *Escherichia coli* extract) to the *T. brucei* cell extracts, as described before [26]. Among the glycolytic intermediates analyzed, F1,6BP and Gly3P accumulated approximately 5-fold and 49-fold more, respectively, in parental cells grown on glycerol as compared to those grown on glucose (Fig 4G). This significant Gly3P accumulation was confirmed by using an enzymatic determination (Fig 4H). It is noteworthy that this Gly3P accumulation persisted in the parental cells incubated with equal amounts of glycerol and glucose, while it was abolished for the $^{RNAi}$GK.ni mutant (Fig 4H). This huge accumulation of Gly3P, due to the large excess of glycosomal GK activity that consumes ATP to produce Gly3P, is probably responsible for the observed reduction of glycosomal ATP level (Fig 4E and 4F).

It may also be considered that the accumulation of intracellular amounts of Gly3P could inhibit HK and prevent G6P production from glucose. To test this hypothesis, HK activity was measured in the presence of Gly3P in $^{RNAi}$GK.i mutant extracts, rather than parental cells, in order to prevent any interference of glycerol metabolism in the assay. Addition of up to 40 mM Gly3P did not significantly affect *in vitro* HK activity (Fig 4I), which is consistent with previously published data [22]. It is also noteworthy that HK activity was not inhibited by adding up to 5 mM F1,6BP (Fig 4I). This shows that the accumulation of Gly3P *per se* or F1,6BP is not responsible for the preference for glycerol over glucose.

## Correlation between high GK/HK activity ratio and glycerol-induced metabolic contest

Glycerol metabolism was also investigated in the PCF of *T. congolense*, a trypanosome species closely related to *T. brucei* that expresses HK and GK in glycosomes and bears a single *GK* copy, instead of 5 copies as in the *T. brucei* genome. GK activity was 5.5-fold lower in *T.*

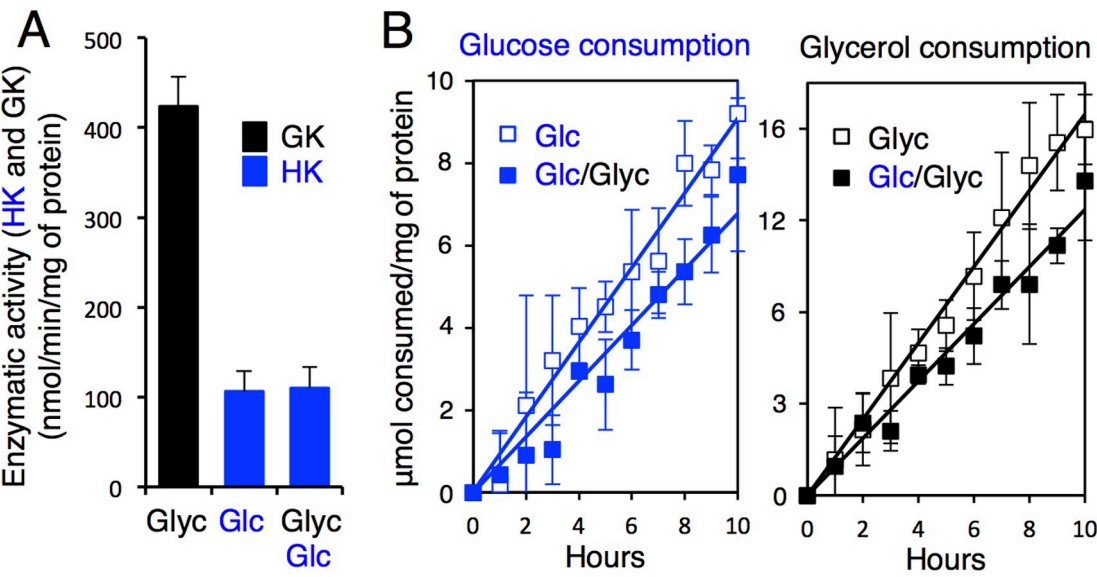

**C**

| | Activity (nmol/min/mg of protein) | | | | Consumption (µmol/h/mg of protein) | | | |
|---|---|---|---|---|---|---|---|---|
| | GK | HK | HK (+Glyc)[a] | Ratio GK/HK[b] | Glc[c] (+Glc)[d] | Glyc[c] (+Glyc)[d] | Glc[c] (+Glc, +Glyc)[d] | Glyc[c] (+Glc, +Glyc)[d] |
| PCF[e] *T. brucei* | 2298 ± 36 | 31.2 ± 0.9 | 2.2 ± 0.3 | 73.7 | 0.955 ± 0.037 | 1.251 ± 0.012 | ND[f] | 1.294 ± 0.050 |
| PCF[g] *T. congolense* | 424 ± 33 | 106 ± 23 | 110 ± 24 | 3.9 | 0.906 ± 0.15 | 1.170 ± 1.13 | 0.744 ± 0.035 | 0.912 ± 0.118 |
| BSF *T. brucei* | 1950 ± 636 | 873 ± 175 | 851 ± 116 | 2.2 | 11.5[h] ± 0.6 | 17.4[h] ± 1.9 | 9.0[h] ± 0.3 | 5.2[h] ± 0.7 |

**Fig 5. Glycerol metabolism in other trypanosomatids.** (A) Glycerol kinase (GK) and hexokinase (HK) activity in total cell extracts of the procyclic form (PCF) of *T. congolense* in the presence of glucose (Glc), glycerol (Glyc) or equal amounts of glucose and glycerol (Glc/Glyc). (B) Consumption of glucose (left) and glycerol (right) by the *T. congolense* PCF incubated in glucose-rich (2 mM), glycerol-rich (2 mM) and glucose/glycerol (2 mM each) conditions. (C) Correlation between high GK/HK activity ratio and preference for glycerol over glucose. BSF, bloodstream form. [a]HK activity in the presence of equimolar amounts of glucose and glycerol. [b]Ratio between GK and HK activity. [c]Rate of glucose (Glc) or glycerol (Glyc) consumption. [d]Culture in the presence of 2 mM glucose (+Glc), 2 mM glycerol (+Glyc) or both (+Glc, +Glyc). [e]Data from Fig 1. [f]ND, not detectable. [g]Data from (B). [h]Data from [20]. Data supporting the results described in this figure can be found at https://zenodo.org/record/5075637#. YORd2B069yA.

*congolense* PCF compared to *T. brucei* PCF, with a GK/HK activity ratio approximately 20-fold lower in *T. congolense* (Fig 5A). Glycerol metabolism did not impair glucose consumption in *T. congolense* PCF as shown by (i) the persistence of high HK activity in the presence of glycerol (Fig 5A) and (ii) the simultaneous consumption of glucose and glycerol (Fig 5B), as observed for the *T. brucei* [RNAi]GK.ni (Fig 1G) cell line. In addition, we previously reported that the BSF of *T. brucei* showed no preference for glycerol or glucose [20]. This is consistent with the unaffected HK activity in the presence of glycerol and with the 28-fold increase of HK activity in the BSF compared to the PCF, while GK activity was equivalent in the 2 forms of the parasite (Fig 5C). Overall, our hypothesis is confirmed by these last data showing a strong

correlation between the preference for glycerol over glucose and the large excess of GK activity compared to HK activity.

In conclusion, we describe here a new mechanism for the regulation of nutrient utilization based on the competition between 2 enzymes (kinases) for a common substrate (ATP). The sequestration of the 2 kinases in the glycosomes is key to this mechanism since these peroxisome-related organelles show limited or no nucleotide exchange with the cytosol on a metabolic timescale [12,28]. Hence, the ATP pool available to glycosomal kinases is limited, offering a situation where significant excess of one kinase (here GK) can abolish almost completely the flux through the other one (HK). The competing kinases (HK and GK) catalyze the first step of their respective pathways and therefore control the utilization of their substrate (glucose and glycerol, respectively). The competition of certain enzymes for a common substrate, in particular at branching metabolic points, is a well-known process used to finely tune metabolic fluxes [29]. However, as far as we know, this is the first example of an almost complete repression of one enzymatic activity (HK) by the large excess of another one (GK) competing for the same substrate, as a mechanism to control nutrient utilization. The consequence of this competition for one substrate, named "metabolic contest," resembles the catabolic repression observed in prokaryotes and yeasts, albeit based on a completely different molecular mechanism, since no nutrient sensing and signaling are required. The advantage of the metabolic contest mechanism over the catabolic repression mechanism is mainly an immediate switch to the less preferred carbon source when the preferred one is exhausted, and with a minimal energy cost since no regulation of gene expression (here HK or GK) is required.

## Materials and Methods

### Trypanosomes and cell cultures

The PCF of *T. brucei* EATRO1125.T7T (TetR-HYG T7RNAPOL-NEO) was cultivated in glucose conditions at 27°C in the presence of 5% $CO_2$ in SDM79 medium containing 10% (v/v) heat-inactivated fetal calf serum and 5 μg/mL hemin [30]. The glycerol-rich/glucose-depleted condition was obtained by replacing glucose with glycerol in SDM79 and adding 50 mM *N*-acetyl-D-glucosamine, which is a non-metabolized glucose analog inhibiting glucose import [31], in order to prevent the consumption of the residual serum-derived glucose (final concentration in the medium: 0.5 mM) [32]. The PCF of *T. congolense* TREU was cultivated at 27°C in the presence of 5% $CO_2$ in TcPCF-3 medium composed of Eagle's Minimum Essential Medium (Sigma-Aldrich) supplemented with 2.2 g/L NaHCO₃, 25 mM HEPES, 0.1 mM hypoxanthine, 2 mM glutamine, 10 mM proline, 20% (v/v) heat-inactivated fetal calf serum and 5 μg/mL hemin [33].

### Production of mutant cell lines

RNAi-mediated inhibition of gene expression of the *GK* genes (Tb927.9.12550–Tb927.9.12630) was performed in the EATRO1125.T7T PCF by expression of stem-loop "sense–antisense" RNA molecules of the targeted sequences [34,35] using the pLew100 expression vector, which contains the phleomycin resistance gene (kindly provided by E. Wirtz and G. Cross) [36]. The sense and antisense version of a 617-bp fragment of the *GK* gene (from position 460 to 1,077) was introduced into the pLew100 vector to produce the pLew-GK-SAS plasmid, as previously described [37]. The EATRO1125.T7T parental cell line was transfected with the NotI-linearized pLew-GK-SAS plasmid in 4-mm electroporation cuvettes with the Gene Pulser Xcell apparatus (Bio-Rad) using the parameters 1,500 V, 25 μF, infinite resistance and 2-pulse mode. Selection of the $^{RNAi}$GK mutant was performed in glucose-rich SDM79 medium containing hygromycin (25 μg/mL), neomycin (10 μg/mL) and phleomycin

(5 μg/mL). Aliquots were frozen in liquid nitrogen to provide stocks of each line that was not cultivated long term in medium. Induction of RNAi cell lines was performed by addition of 1 μg/mL tetracycline.

To generate the constitutive [RNAi]GK cell line ([RNAi]GKcst), the HindIII/BamHI restriction fragment of the pLew-GK-SAS plasmid was inserted into the pLew100 vector missing the tetracycline operator (TetO) sequences required for conditional expression of the downstream cassette. The resulting NotI-linearized pLew-GKcst-SAS plasmid was introduced by electroporation into the EATRO1125.T7T parental cell line before selection with phleomycin (5 μg/mL). The recoded *GK* gene (GKrec) was introduced into the HindIII and BamHI restriction sites of the pHD1336 expression vector (kindly provided by C. Clayton, ZMBH, Heidelberg, Germany) to produce the pHD1336-GKrec plasmid (GeneCust), which was in turn introduced into the [RNAi]GKcst cell line. The resulting [RNAi]GKcst/[OE]GKrec cell line was selected with blasticidin (10 μg/mL), in addition to hygromycin, neomycin, and phleomycin.

To express in glycosomes the ATeam cassette, composed of the ε subunit of the $F_oF_1$-ATP synthase sandwiched by CFP and YFP [24], the full-length *GPDH* gene (Tb927.8.3530), encoding the PTS1-containing glycosomal GPDH, was inserted downstream of the ATeam cassette to generate a gene encoding the glycosomal ATeam-Myc-GPDH recombinant protein. Briefly, a PCR-amplified 1,833-bp fragment containing the ATeam sequence flanked by the HindIII and NdeI restriction sites was inserted into the HindIII and NdeI restriction sites of the pLew100 vector containing the *GPDH* gene preceded by 3 Myc tag sequences. The resulting pLew-ATeam-Myc-GPDH plasmid was introduced in the EATRO1125.T7T parental cell line, and clones were selected with phleomycin (5 μg/mL), in addition to hygromycin and neomycin.

## Western blot analyses

Total protein extracts of *T. brucei* PCF ($5 \times 10^6$ cells) were separated by SDS-PAGE (10%) and immunoblotted on Trans-Blot Turbo Midi PVDF membranes (Bio-Rad) [38]. Immunodetection was performed as previously described [38,39] using as primary antibodies rabbit anti-GK (αGK; 1:2,000; gift from P. A. M. Michels, Edinburgh, UK), rabbit anti-enolase (αENO; 1:100,000; gift from P. A. M. Michels, Edinburgh, UK), rabbit anti-GPDH (αGPDH, 1:100) [40], rabbit anti-PPDK (αPPDK, 1:1,000) [41], rabbit anti-PFR (αPFR; 1:10,000), and mouse anti-Myc 9E10 (αMyc; 1:100; gift from K. Ersfeld, Hull, UK). Anti-rabbit IgG or anti-mouse conjugated to horseradish peroxidase (Bio-Rad, 1:5,000) was used as secondary antibody. Revelation was performed using the Clarity Western ECL Substrate as described by the manufacturer (Bio-Rad). Images were acquired and analyzed with the ImageQuant LAS 4000 luminescent image analyzer.

## Immunofluorescence analyses

Cells were washed twice with PBS and fixed with 4% paraformaldehyde (PFA) for 10 min at room temperature, spread on slides, and permeabilized with 0.05% Triton X-100. After incubation in PBS containing 4% BSA overnight, cells were incubated for 45 min with anti-aldolase rabbit serum (Aldo; 1:1,000; gift from P. Michels, Edinburgh, UK). After washing with PBS, samples were incubated for 45 min with a secondary anti-rabbit IgG antibody conjugated to Alexa Fluor 594 (Thermo Fisher Scientific, Waltham, MA, US). Slides were washed and mounted with SlowFade Gold (Molecular Probes). Images were acquired with MetaMorph software (Molecular Devices, Sunnyvale, CA, US) on a Zeiss Imager Z1 or an Axioplan 2 microscope as previously described [42].

### Fluorescence intensity ratio measurements

The video-microscopy experiments were performed at the Bordeaux Imaging Center (BIC) on an inverted Leica DMI 6000 microscope (Leica Microsystems, Wetzlar, Germany) equipped with a resolutive HQ2 camera (Photometrics, Tucson, AZ, US). The illumination system used was a Lumencor Spectra 7 (Lumencor, Beaverton, OR, US). The objective used was a HCX PL APO CS 63× oil 1.32 NA. This system was controlled by MetaMorph software. Cells were observed in differential interference contrast (DIC) mode by both transmission and fluorescence microscopy (CFP excitation/CFP emission for donor acquisition, CFP excitation/YFP emission for FRET acquisition). We quantified YFP (FRET) and CFP emission using ImageJ software (US National Institutes of Health, Bethesda, MD, US).

### Fluorescence Lifetime Imaging Microscopy (FLIM)

The FLIM measurements were performed at the Bordeaux Imaging Center with the Lambert Instrument FLIM Attachment (LIFA, Lambert Instrument, Roden, Netherlands), which allows the generation of lifetime images using the frequency domain method. This system consists of a modulated intensified CCD camera ($Li^2$ CAM MD), a modulated light excitation light source, and a modulated GenIII image intensifier. For widefield epi-illumination, a modulated LED (light-emitting diode) was used at 451 nm (3 W) for CFP excitation. Both the LED and the intensifier were modulated at frequency up to 100 MHz. A series of 12 images was recorded for each sample. By varying the phase shifts (12 times) between the illuminator and the intensifier modulation, we calculated the phase and modulation for each pixel of the image. Then we determined the sample fluorescence lifetime image using the manufacturer's LI-FLIM software. Lifetimes were referenced to a solution of erythrosin B (1 mg/mL) set at 0.086 ns [43].

### ¹H-NMR spectroscopy experiments

*T. brucei* PCF cells ($3 \times 10^7$ cells/sample) were centrifuged at 1,400*g* for 10 min; then, the pellet was washed twice with PBS, and the cells were incubated for 6 h at 27˚C in 1.5 mL of incubation buffer (PBS supplemented with 5 g/L $NaHCO_3$ [pH 7.4]) with 4 mM [U-$^{13}$C]-glycerol and/or 4 mM glucose. This quantitative ¹H-NMR approach was previously developed to distinguish between [$^{13}$C]-enriched and non-enriched excreted molecules produced from [$^{13}$C]-enriched and non-enriched carbon sources, respectively [14–16]. The viability of the cells during the incubation was checked by microscopic observation. At the end of the incubation, 500 μL of supernatant was collected, and 20 mM maleate was added in this aliquot as internal reference. ¹H-NMR spectra were performed at 125.77 MHz on a Bruker DPX500 spectrometer equipped with a 5-mm broadband probe head. Measurements were recorded at 25˚C with an ERETIC method. This method provides an electronically synthesized reference signal [44]. Acquisition conditions were as follows: 90˚ flip angle, 5,000 Hz spectral width, 32 K memory size, and 9.3 s total recycle time. Measurements were performed with 256 scans, for a total time close to 40 min. Before each experiment, phase of ERETIC peak was precisely adjusted. Resonances of obtained spectra were integrated, and results were expressed relative to ERETIC peak integration.

### Mass spectrometry analyses of ¹³C incorporation into cellular metabolites

For analysis of $^{13}$C incorporation into intracellular metabolites, EATRO1125.T7T parental and mutant cell lines grown in SDM79 medium were washed twice with PBS and resuspended in an incubation solution (PBS containing either 2 mM [U-$^{13}$C]-glycerol or 2 mM [U-$^{13}$C]-

glucose with or without the same amount of unlabeled glucose or glycerol). The cells were incubated for 2 h at 27˚C before being collected on filters by fast filtration and prepared for MS analysis as described before [26]. Total sampling time was below 8 s, and the extraction of intracellular metabolites was carried out by transferring the filters containing the pellets into 5 mL of boiling water for 30 s. The extracts were briefly vortexed (approximately 2 s), immediately filtered (0.2 μm), and chilled with liquid nitrogen. After freeze-drying, the dried extracts were resuspended in 200 μL Milli-Q water prior to analysis. Three replicates were taken from each culture, sampled and analyzed separately. The analyses of metabolites were carried out on a liquid anion exchange chromatography system, the Dionex ICS-5000+ Reagent-Free HPIC System (Thermo Fisher Scientific), coupled to an Q Exactive Plus high-resolution mass spectrometer (Thermo Fisher Scientific), as previously described [45]. Central metabolites were separated within 48 min, using linear gradient elution of KOH applied to an IonPac AS11 column (250 × 2 mm, Dionex) equipped with an AG11 guard column (50 × 2 mm, Dionex) at a flow rate of 0.35 mL/min. The column and autosampler temperature were 30˚C and 4˚C, respectively. Injected sample volume was 15 μL. Mass detection was carried out in negative electrospray ionization (ESI) mode. The settings of the mass spectrometer were as follows: spray voltage 2.75 kV, capillary temperature 325˚C, desolvation temperature 380˚C, and maximum injection time 0.1 s. Nitrogen was used as sheath gas (pressure 50 units) and auxiliary gas (pressure 5 units). The automatic gain control (AGC) was set at 1e6 for full scan mode with a mass resolution of 70,000. Identification of $^{13}$C carbon isotopolog distribution relied on matching accurate masses from Fourier transform mass spectrometry (mass tolerance of 5 ppm) and retention time using TraceFinder 3.2 software. To obtain $^{13}$C labeling patterns ($^{13}$C isotopologs), isotopic clusters were corrected for the natural abundance of isotopes of all elements and for isotopic purity of the tracer, using the in-house software IsoCor, freely available at https://github.com/MetaSys-LISBP/IsoCor, with documentation at https://isocor.readthedocs.io.

## Determination of intracellular metabolite concentrations by IC-HRMS and enzymatic assays

For the quantification of intracellular metabolites, the *T. brucei* PCF EATRO1125.T7T cell line grown in glucose or glycerol conditions was sampled by fast filtration and analyzed as described above. As internal quantification standard, 200 μL of a uniformly [$^{13}$C]-labeled *E. coli* cell extract was added before performing the extraction of intracellular metabolites [46]. The measured concentrations of metabolites were expressed as total cellular concentrations assuming a volume of $10^8$ cells being equal to 5.8 μL [27].

For the enzymatic determination of Gly3P, *T. brucei* PCF cells ($5 \times 10^7$ cells per sample) were washed in PBS and lysed in 100 μL of fresh 0.9 M perchloric acid. After centrifugation at 16,000*g* at 4˚C, 150 μL of $H_2O$ and 75 μL of the KOH (2 M)/MOPS (0.5 M) mix were added to the cellular pellet and incubated for 5 min on ice. After centrifugation at 16,000*g* at 4˚C, the amount of Gly3P contained in the supernatant was determined with the Amplite Fluorimetric G3P Assay Kit, as described by the manufacturer (AAT Bioquest, Euromedex 13837 AAT).

## Determination of glucose and glycerol consumption

To determine the rate of glucose and glycerol consumption, *T. brucei* PCF EATRO1125.T7T (inoculated at $10^7$ cells/mL) or *T. congolense* TREU (inoculated at $5 \times 10^6$ cells/mL) was grown in 10 mL of SDM79 or TcPCF-3 medium, respectively, containing 2.5 mM glucose, 2.5 mM glycerol, or both [30,33]. Aliquots of growth medium (500 μL) were collected periodically during the 10 h of incubation at 27˚C. The quantity of glucose and glycerol present in the medium

was determined using the Glucose GOD-PAP kit (Biolabo, Maizy, France) and the Glycerol Assay Kit (Sigma-Aldrich), respectively. The amount of carbon source consumed at a given time of incubation (Tx) was calculated by subtracting the remaining amounts in the spent medium at Tx from the initial amounts at T0. Then, the rate of glucose and glycerol consumed per h and per mg of protein was calculated from the equation of the linear curve deduced from plotting carbon source consumption as a function of time of incubation. Importantly, we controlled that 100% of the cells remained alive and motile at the end of the 10 h of incubation.

## Enzymatic activities

For enzymatic activities, PCF cells were washed in PBS (10 min, room temperature, 900$g$), resuspended in assay buffer, and after addition of Complete EDTA-free Protease Inhibitor Cocktail (Roche), lysed by sonication (Bioruptor, Diagenode; high intensity, 5–10 cycles, 30 s/30 s on/off). Debris was spun down (15 min, room temperature, 16,000$g$), and the supernatants were used for protein determination with the Pierce BCA Protein Assay Kit in a FLUOstar Omega Plate Reader at 660 nm. For higher throughput and smaller assay volumes, all activity measurements were performed in a 96-well format with a FLUOstar Optima including an automated injection system. Malic enzyme activity was determined as quality control of the cellular extracts, as described before [47]. The baseline reactions were measured for 2 min, and the reactions were started by injection of the specific substrate or a combination of 2 (glucose, glycerol, malate, acetate) for each enzyme. The decrease/increase in absorbance at 350 nm was followed for 3–5 min. The rate was determined from the linear part of the progress curve, and from this the specific activity was calculated. GK activities were determined in 100 mM triethanolamine (pH 7.6), 2.5 mM MgSO$_4$, 10 mM KCl, 0.6 mM ATP, 2 mM phosphoenolpyruvate, 0.6 mM NADH, approximately 1 U lactate dehydrogenase, approximately 1 U pyruvate kinase, and 10 mM glucose and/or glycerol (injected substrate). The buffer for HK measurements contained 100 mM triethanolamine (pH 7.6), 10 mM MgCl$_2$, 0.6 mM ATP, 0.6 mM NADP$^+$, approximately 1 U glucose-6-phosphate dehydrogenase, and 10 mM glucose and/or glycerol (injected substrate) [21]. The same conditions were used for the determination of HK activity in the presence of acetate kinase, except for the addition of 10 mM acetate and recombinant acetate kinase from *E. coli*. For the determination of HK activity, 60 μg of cellular protein was used per well, whereas for GK activity 6 μg per well was used.

## Supporting information

**S1 Fig. Growth curves of the parental (WT) and the tetracycline-induced $^{RNAi}$GK.i cell lines maintained in the presence of 10 mM glucose and glycerol (+Glc, +Glyc), 10 mM glucose (+Glc, −Glyc), 10 mM glycerol (−Glc, +Glyc) or none of them (−Glc, −Glyc).** Cells were maintained in the exponential growth phase (between $10^6$ and $10^7$ cells/mL) and cumulative cell numbers reflect normalization for dilution during cultivation.
(DOCX)

**S2 Fig. HK activity in different combinations (indicated in the table below the graph) of total cell extracts from the parental (WT) and the $^{RNAi}$GK.i cell lines.** The amount of HK remains the same in all samples, while the amount of GK present in the parental sample is diluted with the GK-depleted $^{RNAi}$GK.i sample. The HK and GK activity were determined in the presence of both glucose and glycerol, as performed in the Glc/Glyc conditions (see Fig 2B).
(DOCX)

**S1 Table. Excreted end products from metabolism of [U-$^{13}$C]-glycerol and/or glucose by the parental (EATRO1125.T7T), $^{RNAi}$GK.ni and $^{RNAi}$GK.i procyclic _T. brucei_ cell lines.** The extracellular PBS medium of trypanosomes incubated in the presence of 4 mM of 1 or 2 carbon sources was analyzed by $^1$H-NMR spectroscopy to detect and quantify excreted end products. Data supporting the results described in this table can be found at https://zenodo.org/record/5075637#.YORd2B069yA.
(DOCX)

**S2 Table. Excreted end products from metabolism of [U-$^{13}$C]-glycerol and/or glucose by the parental (EATRO1125.T7T), $^{RNAi}$GKcst, $^{RNAi}$GKcst/$^{OE}$GKrec.ni and $^{RNAi}$GKcst/$^{OE}$GK-rec.i cell lines, grown in the presence of glucose or glycerol (Glyc).** The extracellular PBS medium of trypanosomes incubated in the presence of 4 mM of 1 or 2 carbon sources was analyzed by $^1$H-NMR spectroscopy to detect and quantify excreted end products. Data supporting the results described in this table can be found at https://zenodo.org/record/5075637#.YORd2B069yA.
(DOCX)

## Acknowledgments

We thank Daniel Inaoka (University of Nagasaki, Japan) and Hiromi Imamura (University of Tokyo, Japan) for providing us with the ATeam construct, Paul A. Michels (Edinburgh, UK) for providing us with the anti-GK, anti-aldolase, and anti-enolase immune sera, as well as K. Ersfeld, (Hull University, UK) for the anti-Myc immune serum and Christel Pujol (Bordeaux Imaging Center, France) for her invaluable technical assistance for the FRET and FLIM experiments. The microscopy was done in the Bordeaux Imaging Center, a service unit of CNRS-INSERM and Bordeaux University, member of the national infrastructure France BioImaging.

## Author Contributions

**Conceptualization:** Stefan Allmann, Marion Wargnies, Nicolas Plazolles, Emmanuel Tetaud, Brice Rotureau, Arnaud Mourier, Jean-Charles Portais, Frédéric Bringaud.

**Formal analysis:** Stefan Allmann, Marion Wargnies, Nicolas Plazolles, Edern Cahoreau, Marc Biran, Pauline Morand, Erika Pineda, Hanna Kulyk, Corinne Asencio, Oriana Villafraz.

**Funding acquisition:** Frédéric Bringaud.

**Investigation:** Frédéric Bringaud.

**Methodology:** Stefan Allmann, Marion Wargnies, Nicolas Plazolles, Edern Cahoreau, Marc Biran, Pauline Morand, Erika Pineda, Hanna Kulyk, Corinne Asencio, Oriana Villafraz, Jean-Charles Portais.

**Project administration:** Frédéric Bringaud.

**Supervision:** Loïc Rivière, Emmanuel Tetaud, Brice Rotureau, Arnaud Mourier, Jean-Charles Portais, Frédéric Bringaud.

**Writing – original draft:** Frédéric Bringaud.

**Writing – review & editing:** Nicolas Plazolles, Emmanuel Tetaud, Brice Rotureau, Arnaud Mourier, Jean-Charles Portais.

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
