## [Editor Report · Decision Letter 0]

11 May 2021

Dear Dr Bringaud, 

Thank you for submitting your revised manuscript entitled "'Metabolic contest,' a new way to control carbon source preference" for consideration as a Research Article by PLOS Biology.

Your revisions have now been evaluated by the PLOS Biology editorial staff, and I'm writing to let you know that we would like to send your submission out for re-review.

However, before we can send your manuscript back to the reviewers, we need you to complete your submission by providing the metadata that is required for full assessment. To this end, please login to Editorial Manager where you will find the paper in the 'Submissions Needing Revisions' folder on your homepage. Please click 'Revise Submission' from the Action Links and complete all additional questions in the submission questionnaire.

Please re-submit your manuscript within two working days, i.e. by May 13 2021 11:59PM.

Once your full submission is complete, your paper will undergo a series of checks. Once your manuscript has passed all checks it will be sent back out for re-review. 

Kind regards,

Roli Roberts

Roland Roberts

Senior Editor

PLOS Biology

rroberts@plos.org

---

## [Decision Letter · Decision Letter 1]

24 Jun 2021

Dear Dr Bringaud,

Thank you for submitting your revised Research Article entitled "'Metabolic contest,' a new way to control carbon source preference" for publication in PLOS Biology. I have now obtained advice from one of the original reviewers. Please accept my apologies for the delay while we experienced difficulty contacting the Academic Editor (we finally consulted an alternative Academic Editor).

Based on the review, we will probably accept this manuscript for publication, provided you satisfactorily address the remaining points raised by the reviewers. Please also make sure to address the following data and other policy-related requests.

IMPORTANT:

a) We agree with the reviewer's suggestion about the title, and strongly suggest that you change it to the following slightly modified version: "Glycerol suppresses glucose consumption in Trypanosomes through metabolic contest."

b) Please provide a blurb, according to the instructions in the submission form.

c) Please address my Data Policy requests below; specifically, please supply numerical values underlying Figs 1BCDFGH, 2BCDE, 3ABCD, 4AEFGHI, 5ABC, and cite the location of the data clearly in each relevant Fig legend. 

We expect to receive your revised manuscript within two weeks. 

*Published Peer Review History*

*Early Version*

Sincerely,

Roli Roberts

Senior Editor,

rroberts@plos.org,

PLOS Biology

DATA POLICY:

Regardless of the method selected, please ensure that you provide the individual numerical values that underlie the summary data displayed in the following figure panels as they are essential for readers to assess your analysis and to reproduce it: Figs 1BCDFGH, 2BCDE, 3ABCD, 4AEFGHI, 5ABC. NOTE: the numerical data provided should include all replicates AND the way in which the plotted mean and errors were derived (it should not present only the mean/average values).

We require the original, uncropped and minimally adjusted images supporting all blot and gel results reported in an article's figures or Supporting Information files. We will require these files before a manuscript can be accepted so please prepare and upload them now. Please carefully read our guidelines for how to prepare and upload this data: https://journals.plos.org/plosbiology/s/figures#loc-blot-and-gel-reporting-requirements 

DATA NOT SHOWN?

REVIEWER'S COMMENTS:

Reviewer #2:

Good revision. The only thing the authors may optionally wish to consider is whether a different title may be more clear and impactful, like "Glycerol suppression of glucose consumption in Trypanosomes through metabolic contest"

---

## [Editor Report · Decision Letter 2]

9 Jul 2021

Dear Dr Bringaud,

On behalf of my colleagues and the Academic Editor, Joshua D. Rabinowitz, I'm pleased to say that we can in principle offer to publish your Research Article "Glycerol suppresses glucose consumption in trypanosomes through metabolic contest" in PLOS Biology, provided you address any remaining formatting and reporting issues. These will be detailed in an email that will follow this letter and that you will usually receive within 2-3 business days, during which time no action is required from you. Please note that we will not be able to formally accept your manuscript and schedule it for publication until you have made the required changes.

I should clarify that Dr Rabinowitz was only enlisted as Academic Editor for the final stages; a different Academic Editor helped us with the earlier stages, but was no longer able to handle the manuscript.

PRESS: We frequently collaborate with press offices. If your institution or institutions have a press office, please notify them about your upcoming paper at this point, to enable them to help maximise its impact. If the press office is planning to promote your findings, we would be grateful if they could coordinate with biologypress@plos.org. If you have not yet opted out of the early version process, we ask that you notify us immediately of any press plans so that we may do so on your behalf.

Sincerely,

Roli Roberts 

Roland G Roberts, PhD 

Senior Editor 

PLOS Biology

rroberts@plos.org